behaviour/cognition

domestication, correlated selection responses, tameness, chicken, brain size, fear behaviour

**Author for correspondence:**
Per Jensen
e-mail: per.jensen@liu.se

# Selection for reduced fear in red junglefowl changes brain composition and affects fear memory

## Rebecca Katajamaa and Per Jensen

IFM Biology, AVIAN Behaviour Genomics and Physiology Group, Linköping University, Linköping, Sweden

RK, 0000-0002-1581-8548; PJ, 0000-0001-5491-0649

Brain size reduction is a common trait in domesticated species when compared to wild conspecifics. This reduction can happen through changes in individual brain regions as a response to selection on specific behaviours. We selected red junglefowl for 10 generations for diverging levels of fear towards humans and measured brain size and composition as well as habituation learning and conditioned place preference learning in young chicks. Brain size relative to body size as well as brainstem region size relative to whole brain size were significantly smaller in chicks selected for low fear of humans compared to chicks selected for high fear of humans. However, when including allometric effects in the model, these differences disappear but a tendency towards larger cerebra in low-fear chickens remains. Low-fear line chicks habituated more effectively to a fearful stimulus with prior experience of that same stimulus, whereas high-fear line chicks with previous experience of the stimulus had a response similar to naive chicks. The phenotypical changes are in line with previously described effects of domestication.

## 1. Introduction

Domesticated animals are characterized by a coherent set of phenotypic traits, commonly called the domesticated phenotype [1]. These phenotypic traits involve piebald markings, altered body proportions and increased reproductive capability. Early domestication may have been driven by conscious or unconscious selection for tameness and it has been hypothesized that this may have driven other phenotypic traits through genetic mechanisms such as linkage or pleiotropy [2]. Probably the most well-known study of this was made on silver foxes (*Vulpes vulpes*) that were selected on reduced fear of humans [3]. The phenotype of the silver foxes changed within only a few generations of

selection, inducing traits such as white pigmentation and altered reproductive season. These observed changes correspond to the domesticated phenotype. Similar selection experiments have been done on mink (*Neovison vison*) [4], rats (*Rattus norvegicus*) [5] and by ourselves on chickens (*Gallus gallus*) [6], with similar results.

Reduced brain size compared to wild conspecifics is a common trait in domesticated animals [7]. According to the mosaic evolution hypothesis, changes in brain size can happen through selection on individual regions of the brain as a response to selection on specific behaviours [8]. By contrast, the concerted evolution hypothesis suggests that changes in brain size primarily happen in a concerted fashion on the entire brain [9]. Studies comparing brain size between the ancestral red junglefowl and domesticated white leghorn layers found that the domesticated breed has a larger absolute brain mass, but a smaller brain size relative to body mass [10]. However, cerebellum size is proportionally larger in the domesticated breed, demonstrating a complex genetic architecture for brain proportions, in which brain region sizes vary largely independently of each other. Furthermore, we have previously found that adult relative brain size was reduced by only a few generations of selection for increased tameness in red junglefowl, mirroring the effects found in domesticated chickens [11]. As for domesticates, the cerebrum was smaller in birds selected for low fear, whereas the cerebellum was larger. These results from chickens suggest that the mosaic evolution hypothesis better explains the complex responses to selection during domestication.

Brain size, as well as composition, can be related to cognitive processes. For example, diversity in innovativeness is positively correlated with brain size across different bird species [12]. However, inter-species comparisons in brain size and cognitive abilities are complicated because of other factors that are unique to each species. Associations between brain size and cognitive abilities are probably more interesting in intra-species comparisons. Guppies artificially selected for large or small brains differ in learning efficiency [13], and a recent study found that brains from different dog breeds differ not only based on breed differences in skull shape or body size, but are also correlated to behavioural specializations of different breeds [14].

In the present paper, we report alterations in brain size and composition and the outcome of behaviour tests performed on red junglefowl selected across 10 generations for high or low fear of humans, as a way to mimic the early phases of chicken domestication. The tests were chosen to assess cognitive differences between the selection lines that could be associated with the earlier demonstrated differences in brain size and composition. First, we used a conditioned place preference (CPP) test, which has generally been used to test conditioning effects of different drugs, mainly in rodent models. It is based on classical conditioning where animals learn through associative learning to pair an unconditioned stimulus (e.g. a colour) with a conditioned stimulus (e.g. a food reward, or a painful stimulus) [15]. The CPP test has also been used for assessing conditioning effects of other stimuli relevant from an animal welfare perspective, such as access to a running wheel in rats [16] and aversive stimuli in chickens [17]. Chickens also develop a CPP in response to food [18]. As a second test, we included a simpler form of learning, i.e. non-associative learning where animals change their behaviour in response to a stimulus that has not been previously paired with a reward or punishment. Habituation is such an example of non-associative learning and usually results in a decreased behavioural response to a neutral stimulus over time.

The aim of the present study was to analyse changes in total brain size as well as relative sizes of different brain regions in juvenile red junglefowl selected for diverging levels of fear of human and, secondly, to test possible differences in cognitive functions in selection lines with different brain sizes. We hypothesize that selection for reduced fear of humans will lead to smaller brains and affect the performance in cognitive tasks in chicks through correlated selection responses.

# 2. Material and methods

## 2.1. Animals and housing

We selected red junglefowl for high and low fear of humans for 10 generations. The birds used in this study were from the 9th and 10th selected generation. The origin of our selection lines are chickens from two different zoo populations, Copenhagen zoo and Götala research station. These two populations were used to create the outbred parental generations after having been interbred for two generations. The most and least fearful parental individuals were the starting point of our selection lines.

Fearfulness towards humans was determined in a standardized behavioural test when the birds were 12 weeks old. The reaction to an approaching human was tested individually in an arena measuring $3 \times 1$ m. Selection was done within each selection line after the first outbred parental generation. The most fearful and least fearful individuals from each generation were used as breeders in the high- and low-fear lines, respectively. Agnvall *et al.* [6] contains the full information on the behaviour test used for the selection as well as a more detailed description of the breeding programme. In the following, we refer to the birds from the high-fear selection line as 'high' and from the low selection line as 'low'.

Immediately following hatching, the chicks were individually wing tagged and weighed. For the first one to three weeks, the birds were housed in groups of about 10 birds in small pens ($72 \times 72 \times 57$ cm) containing heaters and littered floors. After that they were moved to larger pens measuring $2 \times 2$ m, the heaters were removed and perches were added. Chicks were provided with *ad libitum* access to feed and water. A 12 : 12 h dark : light period was maintained in the animal housing facility.

## 2.2. Brain tissue collection

Brain tissue was dissected and weighed in the 10th generation of selected red junglefowl in 18 high chicks (10 females, 8 males) and 18 low chicks (13 females, 5 males) when they were five weeks old. After weighing for body weight, the animals were culled by rapid decapitation. Dissection of brain tissue was done immediately after. The whole brain was removed from the skull and subsequently divided into five regions; the cerebrum, optic lobes, cerebellum, hypothalamus and thalamus (in one part), and a brainstem region containing the rest of the midbrain as well as pons and medulla. Each individual part was weighed to assess changes in size between the two different selection lines. Weighing was done immediately after dissection to the nearest 0.01 g (wet mass). In our analysis, we used both the relative and absolute weight of the collected tissue. The relative weight of the whole brain was calculated as tissue weight divided by total body weight of the animal. For each individual brain part, the relative weight was calculated as tissue weight divided by total brain weight.

## 2.3. Fear habituation test

Chicks often respond strongly to a sudden harmless light flash, and this was used to design a fear habituation (FH) test. The FH test was done 8 days after hatch. An arena ($25 \times 25 \times 30$ cm) with a blue LED light (round light source, measuring 48 mm in diameter) in the bottom was used. The bottom of the arena was covered with a transparent plastic film with a corrugated surface structure on it to avoid slippage. From the 9th generation of selection, we tested in total 20 high (10 females and 10 males) and 20 low chicks (9 females, 11 males) twice with 24–28 h in between tests (prior experience group). Furthermore, a group of 10 high (6 females, 4 males) and 9 low (5 females, 4 males) were tested once only (naive group).

### 2.3.1. Procedure

In order to separate short-time habituation from habituation over longer time, the procedure of the test was divided into two phases run over two consecutive days. Day one was used as an experience phase for the prior experience group and the test phase was run on both groups on day two. The same arena was used both days. In the first phase, chickens from the prior experience group were placed in the arena and exposed to five flashes of blue light. The light was flashed for 1 s with 30 s intervals. Thirty seconds after the last light flash, the chick was removed and returned to the home pen. The first light flash occurred 30 s after the chick was placed in the arena.

The test on day two was carried out 24–28 h after the prior experience phase. In the test, the chicks went through a procedure similar to the experience phase. Four chicks were tested at the same time in four different arenas. They were placed in the arenas with the lights turned off. Thirty seconds after the lights had been turned on, the chicks were exposed to the first blue light flash. A total of 10 light flashes were used with 30 s between them. The birds were removed from the arenas 30 s after the last light flash. The naive group only went through the test procedure on day two and had consequently no prior experience of the stimulus.

Test sessions were video recorded and subsequently analysed using The Observer® XT version 13 (Noldus Inc.). Each chick's fear reaction to each light cue was recorded according to a predefined scale (table 1). All recordings were done by the same observer (the first author). A selection of 10 birds were resampled by the last author, blind to selection lines, in order to check the consistency of the recordings. The correlation between the observer scores was $r_s = 0.80$, $p < 0.001$.

**Table 1.** Ethogram describing the different fear reaction levels identified in the fear habituation test.

| fear reaction | description |
| --- | --- |
| 1 | no reaction |
| 2 | mild reaction; turns head, takes a few steps, no rushed movements |
| 3 | startled; runs/walks fast one length of arena, reaction delayed approximately 1 s. Escape attempt. None or short duration of wing flapping. More than 1 length of arena if not hitting wall |
| 4 | very startled; wing flapping and running into walls. Moves 2 or more lengths of arena. Stops briefly when hitting wall before continuing |
| 5 | extremely startled; rapid, prolonged wing flapping, repeatedly running into walls, 3 or more lengths of the arena moved |
| 6 | freeze |

## 2.4. Conditioned place preference

The second test was a CPP test, performed 8 days after hatch. We used an arena with two compartments which the birds could freely move between through an opening ($13.5 \times 15$ cm). Each compartment was covered with a black and white pattern around the walls, either black dots or black stripes on a white background. The bottom of the arena was covered with a transparent plastic film similar to the set-up in the FH test to avoid slippage. Each compartment in the arena measured $28 \times 34 \times 40$ cm. We used divergently fear-selected red junglefowl from the 10th generation, in total 15 high chicks (8 females, 7 males) and 16 low chicks (12 females, 4 males) for the comparison. Half of the chicks were conditioned on reward in the striped compartment and the other half was conditioned on reward in the dotted compartment. Compartment pattern indicating reward was evenly distributed with respect to selection line. Test sessions were video recorded and analysed using The Observer® XT version 13 (Noldus). We measured the total time spent in each compartment as well as first choice.

### 2.4.1. Procedure

Mealworms were used as the reward in the conditioned compartment. The conditioning phase consisted of five exposures to each compartment of the CPP arena. Mealworms were provided in the rewarded compartment only. Each chick was placed in one end of the compartment, with a dish containing mealworms (reward) or an empty dish (unrewarded). A mirror was placed behind the dish (diameter 7 cm, height 2.5 cm) in both compartments during conditioning to mimic a social companion and thereby reduce stress from social isolation. The chicks were allowed to explore the arena for 30 s in each exposure. After five exposures in each arena they were moved back to the home pen. We opted for a short duration of training as we wanted to create a possible contrast in learning outcome.

Between 24 and 28 h after conditioning, the chicks went through the testing phase, during which there was no reward provided. The arena was equipped with a triangular start box ($13 \times 13 \times 13$ cm, height 19 cm) with a door ($12 \times 15$ cm) that allowed the chicks to enter either of the two compartments through the side. The door was covered with wire mesh to allow the chicks to inspect the compartments of the arena before entering. No mirrors were used in the test session. We allowed the chicks a total time of 30 s for inspecting the arena compartments before removing the wire mesh and letting them enter either of the compartments. The wire mesh was replaced after the chicks had entered one of the compartments of the arena. This test consisted of a 5 min test time in which the chicks were allowed to explore and move freely between both compartments. After 5 min, the chicks were removed and returned to their home pen. We decided to use only 5 min for the test sessions since young chicks are very sensitive to social isolation.

## 2.5. Statistical analysis

Statistical analysis on all measurements related to brain size, body size and CPP test were done using the generalized linear model with scale response 'gamma' with link function 'log' in SPSS v. 24.0. In the model, selection (high versus low) and sex as well as the interaction between them were used as predictors. Interaction effects were removed from the model when they were not significant.

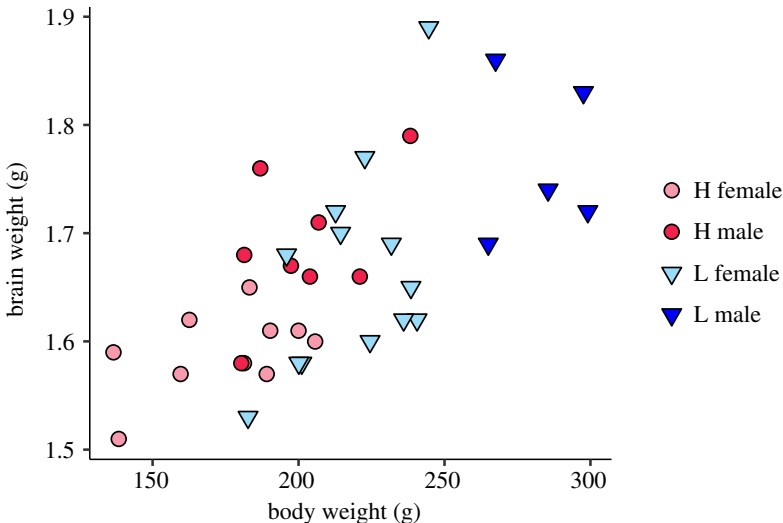

**Figure 1.** Absolute brain size (g) plotted against body weight (g) in red junglefowl of the 10th generation selected for high and low fear of humans.

**Table 2.** Results for significance tests (Wald $\chi^2$-tests) performed on generalized linear model for absolute brain and brain region weights.

| brain region | model term | $\chi^2$ | $p$-value |
|---|---|---|---|
| cerebrum | line | 9.84 | 0.002* |
| | sex | 11.40 | 0.001* |
| | line × sex | 0.020 | 0.888 |
| brainstem region | line | 0.317 | 0.573 |
| | sex | 1.98 | 0.159 |
| | line × sex | 1.81 | 0.179 |
| optic lobes | line | 8.31 | 0.004* |
| | sex | 13.3 | <0.001* |
| | line × sex | 2.19 | 0.139 |
| cerebellum | line | 6.23 | 0.013* |
| | sex | 11.8 | 0.001* |
| | line × sex | 0.808 | 0.369 |
| hypothalamus/thalamus | line | 2.29 | 0.130 |
| | sex | 1.04 | 0.308 |
| | line × sex | 1.72 | 0.190 |
| total brain | line | 10.6 | 0.001* |
| | sex | 17.4 | <0.001* |
| | line × sex | 0.001 | 0.970 |

*Significant effect ($p < 0.05$).
**Tendency towards significance ($0.05 < p < 0.10$).

We first analysed the effects of selection on brain and brain region mass by the previously mentioned model, without taking body size into account. The analysis was done on total brain weight and brain region weight as well as on the relative brain and relative brain region weights. Relative brain weight was calculated by dividing total brain weight by the body weight to get a percentage of total body weight composed of brain tissue. Relative brain region weight was similarly calculated by dividing the region weight by total brain weight to obtain a measure of the percentage of total brain weight

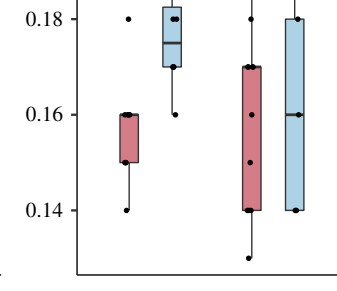
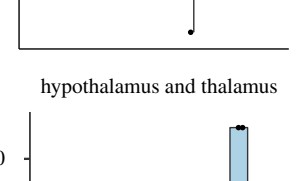
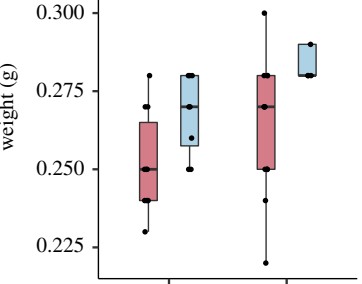
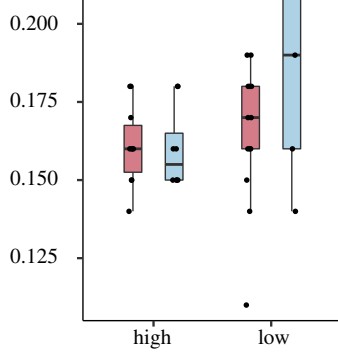
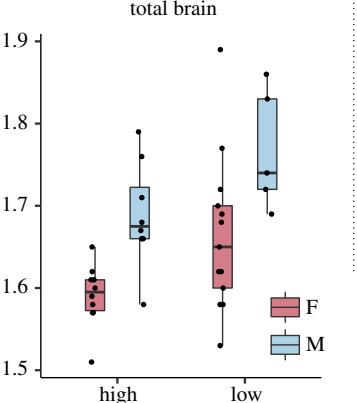

**Figure 2.** Absolute weight (g) of total brain and brain regions in red junglefowl of the 10th generation selected for high and low fear of humans.

composed of each region. In order to take allometric effects into account, we did a further analysis on effects of selection on total brain size by adding body weight and weight of rest of brain (ROB; the weight of the brain except the part being analysed) as covariates to the model. Body weight was used as a covariate in the model for absolute brain size and ROB was used as a covariate in the model for each brain region.

The FH test was tested with a cumulative link mixed model using the package ordinal in R v. 3.5.2. Selection line and time as well as their interaction were used as predictors, and animal identity was included as random effect. Plots were built using R v. 3.5.2 and package ggplot2.

## 3. Results

The body weight of the low-fear line was significantly higher than of the high-fear birds, and females were significantly smaller than males (effects of selection: $\chi^2 = 57.4$, $p < 0.001$; sex: $\chi^2 = 28.9$, $p < 0.001$). The absolute brain size was significantly correlated with body size ($R_S = 0.671$, $p < 0.001$; figure 1), and absolute brain size was significantly larger in the low-fear line compared to the high-fear line and in males compared to females (table 2 and figure 2). Absolute sizes of the cerebrum, optic lobes and cerebellum were larger in the low-fear line, with sex effects in all three, where males had larger regions (table 2 and figure 2). No interaction effects were found in any of the tested regions.

The per cent of total body weight that was composed of brain was significantly smaller in the low line and in males compared to females, with no significant interaction effects between sex and selection line (table 3 and figure 3). Comparing the separate brain regions, the brainstem region constituted a significantly smaller proportion of the total brain mass in low-fear chickens compared to high-fear chickens (table 3 and figure 3), while there were no differences in this respect for any of the other brain regions. No effects of sex or interaction effects were found in this case.

When taking allometric effects into consideration, absolute brain weight was not significantly affected by selection and neither were weights of cerebellum or optic lobes, whereas there was a tendency towards an effect in cerebrum weight where the cerebra of low-fear chickens were larger (table 4 and figure 2). We

**Table 3.** Results for significance tests (Wald $\chi^2$-tests) performed on generalized linear model for relative brain and brain region weights.

| brain region | model term | $\chi^2$ | p-value |
|---|---|---|---|
| cerebrum | line | 0.114 | 0.735 |
| | sex | 0.685 | 0.408 |
| | line × sex | 0.045 | 0.832 |
| brainstem region | line | 4.13 | 0.042* |
| | sex | 0.043 | 0.837 |
| | line × sex | 2.25 | 0.134 |
| optic lobes | line | 0.234 | 0.629 |
| | sex | 1.28 | 0.258 |
| | line × sex | 3.58 | 0.058** |
| cerebellum | line | 0.356 | 0.551 |
| | sex | 1.14 | 0.285 |
| | line × sex | 1.37 | 0.243 |
| hypothalamus/thalamus | line | 0.142 | 0.706 |
| | sex | 0.275 | 0.600 |
| | line × sex | 1.86 | 0.173 |
| total brain | line | 49.5 | <0.001* |
| | sex | 17.5 | <0.001* |
| | line × sex | 2.43 | 0.119 |

*Significant effect ($p < 0.05$).
**Tendency towards significance ($0.05 < p < 0.10$).

found no interaction effects, but there was a tendency towards an interaction between sex and selection line in optic lobes weight.

In the FH test, birds from both selection lines reacted with a strong startle response to the first flash of light and showed a similar speed of habituation when they had no prior experience of the fearful stimulus (effects of selection: $z = -2.608$, $p < 0.010$; time: $z = -9.535$, $p < 0.001$; interaction selection × time: $z = -1.649$, $p > 0.05$; figure 4a). However, there was a significant effect of selection line as well as time (exposure numbers) on habituation test in the group with prior experience (effects of selection: $z = 0.453$, $p > 0.05$; time: $z = -6.782$, $p < 0.001$; interaction selection × time: $z = -1.845$, $p > 0.05$; figure 4b). The degree of fear reaction declined with repeated exposures in both selection lines when the birds had no prior experience. On the second occasion, low-fear chickens reacted less to the light throughout the test, whereas the fear reaction declined in both lines over time.

In the CPP test, no effects were found on time spent in the rewarded compartment (effects of selection: $\chi^2 = 1.3$, $p > 0.05$; sex: $\chi^2 = 1.2$, $p > 0.05$; interaction selection × sex: $\chi^2 = 0.081$, $p > 0.05$) or on first choice of compartment in the CPP-test (effects of selection: $\chi^2 = 0.32$, $p > 0.05$; sex: $\chi^2 = 0.32$, $p > 0.05$; interaction selection × sex: $\chi^2 = 0.32$, $p > 0.05$).

# 4. Discussion

Our results show that while total brain mass increased allometrically with body size in response to selection for low fear of humans in red junglefowl, the per cent of body weight composed of brain actually decreased, corroborating earlier findings in adult birds [11]. The size changes closely resemble those seen in domesticated chickens compared to their ancestors [10]. Taking body size into account in the generalized linear model, in order to control for allometric effects, eliminated the effect of the selection on brain size. However, even when controlling for allometric effects, by including ROB in the model, low-fear birds tended to have a larger cerebrum. Furthermore, the percentage of the brain consisting of the brainstem region was smaller in the low-fear birds. We also found that the habituation

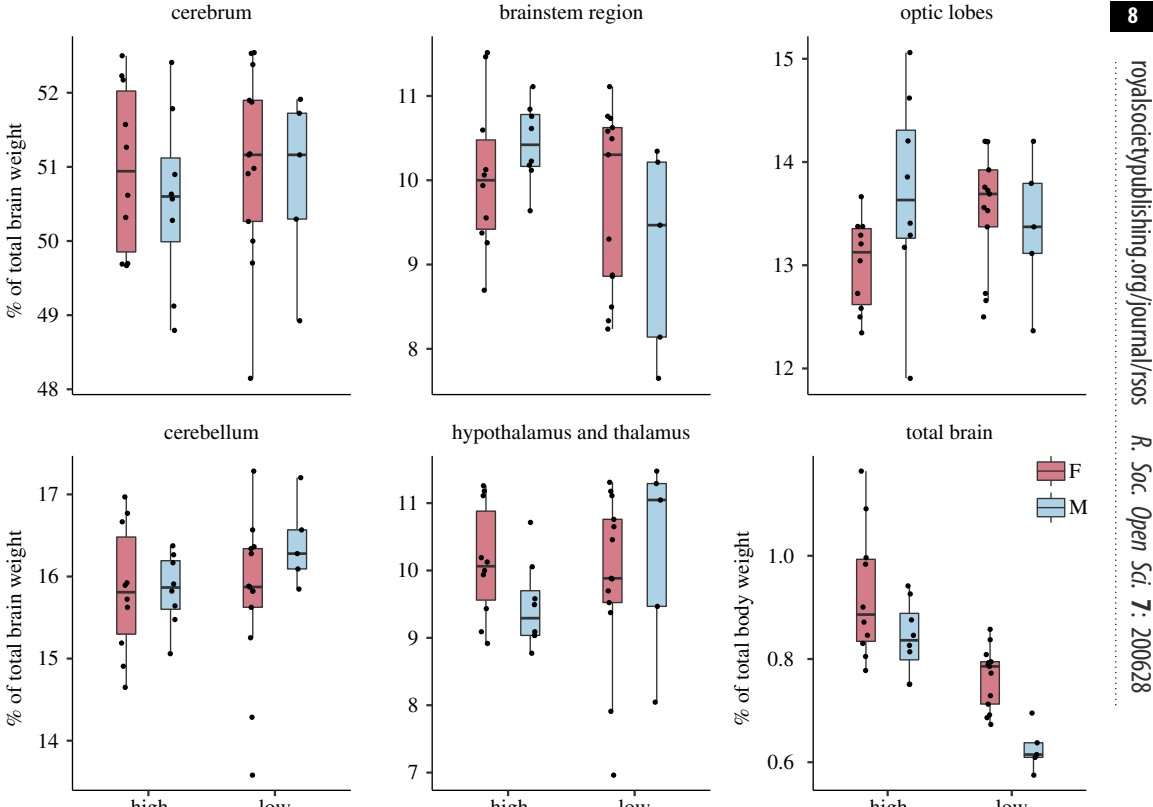

**Figure 3.** Proportion (%) of total brain composed of each region and proportion (%) of whole body weight composed of brain weight in red junglefowl of the 10th generation selected for high and low fear of humans.

to a frightening stimulus was affected by selection, in that birds from the low-fear line reacted less and habituated faster when having prior experience of the stimulus. No effects were found in the CPP test, indicating that selection on level of fear of humans may not have affected associative learning as measured by this test. Our results do not link the behavioural differences to the brain size variation, but it remains a possibility for future research that such a link exists.

It has previously been shown that although absolute brain size, and cerebellum size in particular, is larger in domesticated white leghorn in comparison to ancestral red junglefowl, the proportion of brain out of total body mass is smaller, indicating complex effects of domestication [10]. Other studies have also shown that selection for various traits can cause secondary effects on brain size. Japanese quail (*Coturnix japonica*) selected for high prenatal maternal investment had a smaller cerebellum compared to individuals selected for low maternal investment in a study where egg mass was used as the selection criterion [19]. The white leghorn is a layer breed that lays around 300 eggs per year, each considerably larger than those of the ancestral red junglefowl. The brain is an energetically costly organ [20,21] and therefore there may be trade-offs between brain size and energetically costly phenotypes.

We have previously found that red junglefowl selected for reduced fear of humans show several traits associated with domesticated chickens. They have larger offspring, produce larger eggs [22] and have a smaller brain size with a smaller cerebrum, but relatively larger cerebellum as adults [11]. Hence, it is clear that specific regions of the brain can change relatively independently in size in response to selection on other and apparently unrelated traits, possibly as an adaptive response to the trade-off of increased reproductive efforts. A study on the genetics of brain size, using an intercross between red junglefowl and white leghorns, found that different brain regions are controlled by different genetic architectures, further supporting that brain regions can change in size independently of one another [10]. The brain's proportion of the body weight in the young birds tested in this study is reduced in the low-fear line, and the proportional composition of the brain also differed between the selection lines. Adding body weight and ROB as a covariate to the analysis in order to control for allometric effects reveals a somewhat different picture, where there is a tendency towards an effect on cerebrum size, similar to that seen in adult birds in a previous generation. Regardless of allometric effects, our results clearly show that brain size and composition can change in response to selection for a behavioural trait.

**Table 4.** Results for significance tests (Wald $\chi^2$-tests) performed on generalized linear model for absolute brain and brain region weights with body weight (total brain) or ROB (brain regions) as covariates.

| brain region | model term | $\chi^2$ | p-value |
|---|---|---|---|
| cerebrum | line | 24.63 | 0.055** |
| | sex | 1.81 | 0.18 |
| | ROB | 13.88 | <0.001* |
| | line × sex | 0.035 | 0.85 |
| brainstem region | line | 1.38 | 0.24 |
| | sex | 0.23 | 0.63 |
| | ROB | 1.70 | 0.19 |
| | line × sex | 2.16 | 0.14 |
| optic lobes | line | 0.93 | 0.33 |
| | sex | 2.84 | 0.092** |
| | ROB | 11.60 | 0.001* |
| | line × sex | 3.60 | 0.058** |
| cerebellum | line | 1.53 | 0.22 |
| | sex | 2.36 | 0.13 |
| | ROB | 13.34 | <0.001* |
| | line × sex | 1.31 | 0.25 |
| hypothalamus/thalamus | line | 1.08 | 0.30 |
| | sex | 0.19 | 0.66 |
| | ROB | 0.48 | 0.49 |
| | line × sex | 1.83 | 0.18 |
| total brain | line | 0.011 | 0.92 |
| | sex | 2.19 | 0.14 |
| | body weight | 6.50 | 0.011* |
| | line × sex | 1.17 | 0.28 |

*Significant effect ($p < 0.05$).
**Tendency towards significance ($0.05 < p < 0.10$).

The chicken is a precocial species and therefore the brain is much more developed at hatch compared to altricial species. Post-hatch brain growth in white leghorns and unselected red junglefowl is largely consistent, even when divided into different brain regions [10]. However, the same study found that relative brain region growth (% of total brain mass) has different trajectories depending on region. This can explain why we did not find exactly the same differences between selection lines in brain region sizes in young birds as we have previously reported in older birds [11].

We hypothesized based on previous findings on changes in brain size of adult birds in our selection lines [11] that there may have been some side effects on memory or learning. While we found no effects in our CPP test, there was a clear difference in the FH test, supporting the assumption that selection for tameness also affects cognitive processes, regardless of whether there is a link between brain region size and cognition or not. It should be noted that we have no evidence that the behavioural differences between the selection lines are in any way related to the variations we have found in brain size. However, previous studies have found correlations between brain size and behaviour that are similar to our findings [19,23], making future research on the subject a possibility for our selection lines.

The FH test showed that there was a clear effect on how the birds habituated to a fearful stimulus. The low selection line responded less than the high when they had prior experience of a fearful stimulus. This indicates a more efficient habituation capacity towards fearful stimuli in the low line. A challenge for domesticated animals is living in an artificial environment created by humans, with many potentially frightening but harmless stimuli such as sudden loud noises and the presence of humans. Whereas

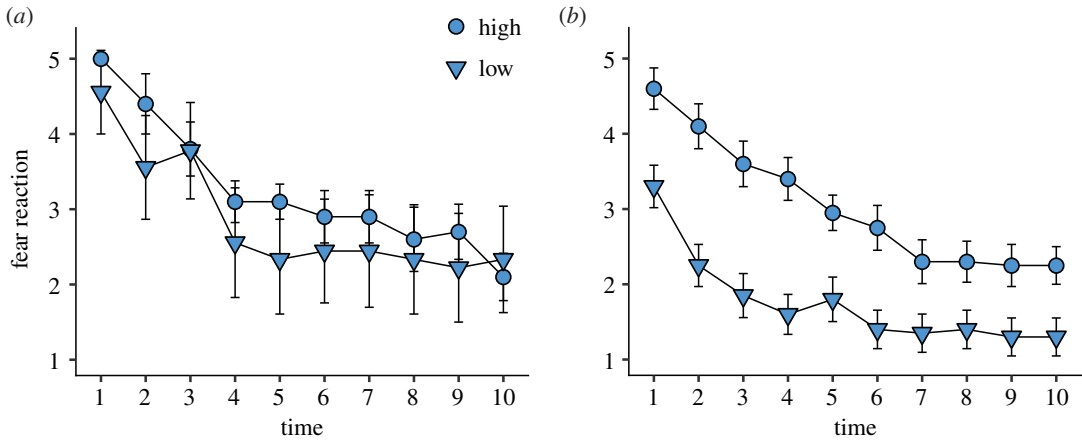

**Figure 4.** Behavioural reaction over time in FH test in the 9th generation red junglefowl selected for high and low fear of humans. (*a*) Naive group. (*b*) Prior experience group.

animals in captivity will benefit from habituating to these circumstances and being less fearful, animals in the wild may not afford loss of vigilance in order to avoid predation. Decreased fearfulness, a common trait in domesticated animals, is associated with an attenuation of hypothalamic-pituitary-adrenal axis reactivity [24]. Selection on a lower level of fear of humans seems to not generalize to other types of fear in our population of chickens [25], but it seems that the selection on tameness has affected habituation.

We found no effects of selection line on the performance in the CPP test, indicating that selection on fear level towards humans does not have a general effect on associative learning. However, it should be noted that there are different protocols for this test, and we may not have used the optimal one to detect any differences. The time recommended for the conditioning session in chickens is 25–30 min [18]. In our tests, we used a very short time duration for the conditioning session in comparison to the recommendation. The intention behind that was to create a possible contrast in learning outcome and therefore the goal was that not all the birds should learn the connection between food reward and the compartment pattern. Whereas the most straightforward interpretation of the results is that the selection lines did not differ in their ability to learn a place preference, it is possible that our test would have benefited from a longer conditioning session or repeated sessions over a day or more.

One relevant issue in this study is the study population. We are aware that there is a lack of replication in the traditional sense, where several independent selection lines is the preferred design. Without replicated selection lines, changes occurring over generations may be due to genetic drift as a result of limited genetic variation in the founders. However, the parental generation used was derived from systematic outbreeding based on two unrelated populations for two generations before carrying out the selection regime, which increased the genetic variation among the parental individuals [6]. Our results from this selection experiment are therefore at least indicative of the possible correlated selection responses occurring with selection on a single behavioural trait, level of fear of humans.

In conclusion, we found that proportion of brain weight relative to total body weight, and proportion of brainstem region size relative to total brain size, both decreased as correlated responses to selection for increased tameness in red junglefowl. This difference in brain size is largely an allometric response to increase in body size in the low-fear line. We also found that FH was more effective in the low-fear line after previous experience with the frightening stimulus, indicating an effect of the selection on habituation learning. Further research is required on possible effects on other cognitive processes in response to selection on tameness, and to ascertain any functional connection between behaviour and brain size and morphology. The phenotypical changes are in line with what is normally found in the domesticated phenotype, i.e. less fearful animals and relatively smaller brains, indicating that reduced fear of humans may to some extent have driven several of the changes incurred by domestication.

Ethics. All experimental protocols were approved by Linköping Council for Ethical Licencing of Animal Experiments, ethical permit no. 50–13. Experiments were carried out in accordance with the approved guidelines.
Data accessibility. The datasets supporting this article have been uploaded as part of the electronic supplementary material.

Authors' contributions. Conception, design, interpretation of data and draft of the manuscript written and prepared by R.K. and P.J. Data acquisition and analysis by R.K. Both authors gave final approval for publication.

Competing interests. We declare we have no competing interests.

Funding. Funding was obtained from a grant to P.J. from the Swedish Research Council (grant no. 2015-05444).

Acknowledgements. We thank our technicians Julia Buskas and Enya van Poucke for taking great care of the animals and for assistance with experiments, Caroline Lindholm and Laura Garnham for assistance with dissections, and Martin Brengdahl for help with part of the statistical analysis.

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
