## [Reviewer comments · Royal Society Open Science]

Review History

RSOS-191559.R0 (Original submission)

Review form: Reviewer 1

Is the manuscript scientifically sound in its present form?

No

Are the interpretations and conclusions justified by the results?

No

Is the language acceptable?

Yes

Do you have any ethical concerns with this paper?

No

Have you any concerns about statistical analyses in this paper?

Yes

Recommendation?

Major revision is needed (please make suggestions in comments)

Comments to the Author(s)

Katajamaa and Jensen's study test if fear memory, brain size and to some extent brain region sizes differ between two lines of Red Junglefowl selected for divergent fear responses. They conclude that there are differences in fear memory between the two selection lines as well as a difference in relative brain size. Their results parallel the domestication process and potentially have implications for understanding how selection for reduced fear has consequences for behaviour and brain. However, I found the Introduction and Discussion to be somewhat disorganized, some of the statistical details are missing and the brain data does not support their conclusions when analyzed with more appropriate methods. I provide more details comments below.

1. At a general level, the Introduction lacked focus and many of the paragraphs were not linked well to one another making it difficult to determine what the overall goal of the study was. I think the authors would be better served by making the paragraph on lines 80-94 as the beginning of their Introduction and reorganize the rest of it around the central idea of how domestication affects the brain and behaviour. I would also recommend de-emphasizing the putative link between cognition and brain size. Yes, there are many studies that support such a relationship, but there are many others that do not. Further, in the context of the current study, it is unclear whether the change in brain and midbrain sizes are related to fear memory or if these are two phenotypic changes that co-occur with the selection experiment. That is, selection for divergent fear responses has resulted in changes in relative midbrain size and fear memory, but the two traits might not be directly related to one another.
2. Also missing from the end of the Introduction was the general hypothesis being tested and/or predictions.
3. There are a few aspects of the brain measurements that require some clarification. First, was the tissue all weighed fresh and not fixed? Second, was the hypothalamus dissected from the whole brain? Given the size and location of the hypothalamus, my concern is whether it could be accurately dissected from fresh brain tissue, even with the assistance of a dissecting microscope. In examining their data file, the mass of the hypothalamus is similar to that of the midbrain, indicating to me that this is not a measurement of hypothalamus. Is it possible that these terms are meant to be tegmentum and tectum? Third, analyzing ratios in the study of relative brain size and brain region sizes is problematic for a number of reasons, not the least of which is that the ratios generated are often correlated with the denominator and therefore do not effectively remove allometric effects. The preferred method for such analyses is an analysis of covariance or linear mixed model in which body size is a covariate for brain size and brain size (minus that of the region of interest) is the covariate for brain region size. In my quick analysis of the authors data file, an ANCOVA of brain size did not yield any significant differences between the two lines and in the scatterplot, there is substantial overlap between the two lines in Cartesian space. There is also no significant difference in relative midbrain size when analyzed in an ANCOVA. I therefore urge the authors to reconsider their statistical methods for analyzing the brain data.
4. In the Results section, statistics are only reported for the CPP test. The authors need to include statistical information for all other tests that were run. These could be provided in the text or a table (or even an appendix), but need to be included.
5. The brain measurements should not be reported in bar graphs. The values themselves cannot be zero so box and whisker plots would be more appropriate. Further, the use of an ANCOVA or linear mixed model necessitates the use of scatterplots for the brain data to reflect both the statistical test and provide the reader with critically important information on the distribution of data for both selection lines. In fact, without these tests and changes to the graphs, I am not entirely convinced that there are major differences in the brain of the two selection lines simply because there is not enough data or information presented.

6. The Discussion lacked focus in a similar fashion to the Introduction. Because the fear memory differences might not be directly related to the brain differences, I suggest separating those two components in the Discussion. The paragraph on lines 290-296 is also not necessary. As far as the brain regions are concerned, the authors highlight that differences in cerebellum size are found in other studies, but do not discuss why they did not observe a similar difference here. Also missing was why midbrain would be affected. Did midbrain include optic tectum or just tegmentum? Is it possible that there is a difference in the visual system if optic tectum was included in the midbrain measurement? What brain regions are likely involved in their behavioural tests and are they located in the midbrain or elsewhere? Addressing these questions would allow the authors to discuss the brain data in a more focused way that speaks to their data.

7. Other minor comments:

- a. Lines 18-19. This sentence is a bit vague and is not contributing much to the abstract.
- b. Line 21. Replace "collected samples" with "measured"
- c. Line 42. Delete "in fact" and replace "birds" with "chickens" as this has only been shown in chickens/junglefowl and in other domesticated birds the cerebellum becomes smaller (turkey, pigeon, duck, goose)
- d. Lines 52-53. Delete "on one hand" and "on the other"
- e. Line 59. The cited paper on dogs does not actually show that breeds with larger brains perform better because there is a mismatch between poorly collected brain data from the literature and the behavioural testing. In short, this citation in Animal Cognition is not a good paper.
- f. Line 61. Delete "some"
- g. Line 73. Delete "have" and "been shown to"
- h. Line 262. Delete "of"
- i. Line 274. Replace "has been found to be" with "is"

Review form: Reviewer 2

Is the manuscript scientifically sound in its present form?

Yes

Are the interpretations and conclusions justified by the results?

No

Is the language acceptable?

Yes

Do you have any ethical concerns with this paper?

No

Have you any concerns about statistical analyses in this paper?

Yes

Recommendation?

Reject

Comments to the Author(s)

In this manuscript Katajamaa & Jensen test the effect of selection on reduced fear on brain anatomy and cognitive abilities in Red Junglefowl. They find that high and low-fear populations differ in brain anatomy and some aspects of cognition. My strongest reservation against this manuscript is the lack of replication in the selection set up. Two small populations separated from a larger population will diverge in various traits over generations simply due to genetic

drift. So it is not possible to conclude whether the differences between two populations are due to the selective regime, or due to drift. For this reason artificial selection experiments in evolutionary biology are usually run in replicates, i.e. several up and several down-selected populations. Ideally also several control lines. This has not been done here. This means we cannot be sure that the differences between the lines are really due to selection on fearfulness – although it seems likely, as the results ‘make sense’ when conceptually placing them in the field of domestication, but that may not be enough. It is unfortunately not uncommon that selection lines (especially if done in vertebrates) are not replicated and I am aware that not all see this lack of replication as big problem. A potential way of publishing these data nevertheless (considering the journal) may be to add a prominent disclaimer to the discussion section in which the authors acknowledge that there is no replication but that the results are at least indicative. In case the editor decides that non-replicated studies also have merit and can be published here are some additional comments:

Body size: What is the difference in body size between the line? It seems like there should be strong differences as the results for absolute and relative brain sizes are so different. It is always tricky to compare brains between differently sized animals as the brain and the body often have different allometries and if there is a relative brain size difference this may be due to a genuine change in brain size, or simply a shift in body size with the brains staying constant. What is going on is tricky to disentangle. To get an idea of the type of body/brain size allometries going on here it would be very helpful to include a scatterplot of body size vs. brain size.

Habituation: Is habituation to a fear-eliciting stimulus ‘cognitive ability’? This needs to be explained and justified.

Abstract: The second, third and fourth sentence seem unconnected. Please consider rephrasing so that it is clear how one leads to the next.

Line 61: Why? What is the justification of the research presented here? It would be good to set the stage first for the “Her we test...”. For instance, I suggest putting the paragraph starting line 80 before the previous one.

Line 67: Please add references.

Line 132: This seems like a rather limited sample size (especially for the males). Of course this may be perfectly fine but for that to judge it would be great to see the raw data. I strongly suggest changing the figures 1 and 2 so that the individual data points are visible.

Line 141: If I understand the methods correctly – when computing relative brain part size the weight of the respective brain part is included in the total brain weight? This means that if brain part is divided through total brain, what is above and below the line is not independent from each other’s. What is therefore usually done is to subtract the brain part from total brain (=rest of brain) and then divide brain part by rest of brain. Often this does not qualitatively change the results, I would nevertheless recommend doing it that (more correct) way.

Line 284: If the brains of those lines have already been investigated, why do it again? And why did the previous study find differences in brain regions that are different to the ones reported here? This is important, please clarify.

Line 292: Ref 25 is not suitable here as it is an intra-species study.

Decision letter (RSOS-191559.R0)

17-Oct-2019

Dear Professor Katajamaa:

Manuscript ID RSOS-191559 entitled "Selection for reduced fear in Red Junglefowl reduces brain size and affects fear memory" which you submitted to Royal Society Open Science, has been reviewed. The comments from reviewers are included at the bottom of this letter.

In view of the criticisms of the reviewers, the manuscript has been rejected in its current form. However, a new manuscript may be submitted which takes into consideration these comments.

Please note that resubmitting your manuscript does not guarantee eventual acceptance, and that your resubmission will be subject to peer review before a decision is made.

Your resubmitted manuscript should be submitted by 15-Apr-2020. If you are unable to submit by this date please contact the Editorial Office.

Kind regards,
Anita Kristiansen
Editorial Coordinator
Royal Society Open Science
openscience@royalsociety.org

on behalf of Dr Ryan Y Wong (Associate Editor) and Kevin Padian (Subject Editor)
openscience@royalsociety.org

Subject Editor Comments to Author (Professor Kevin Padian):

Comments to the Author:

Thank you for your very interesting manuscript. Both reviewers like the concept but have concerns, especially about the analysis. They suggest more appropriate statistical tests, and it will be interesting to see if you discover significant differences between groups with re-analysis. We are logging a decision of "reject, resubmit" in order to give you more time than the window afforded for our "major revision" window. Besides, you might find different results and in that case overhaul the manuscript considerably. In any case we wish you the best for a revision.

Associate Editor Comments to Author (Dr Ryan Y Wong):

Comments to the Author:

Dear Rebecca Katajamaa,

Your manuscript has now been reviewed by two reviewers. As can be seen with the appended reviews, both reviewers brought up a number of concerns of the manuscript in its current state. I highlight the comments regarding insufficient methodological details for accurate evaluation,

need for more in depth statistical analyses, interpretation of results and corresponding limitations, in particular. From my own reading of your manuscript and of the reviewer's reports, I acknowledge the considerable amount of revision that may be necessary. Therefore, I recommend a major revision before considering the manuscript further.

Reviewers' Comments to Author:

Reviewer: 1

Comments to the Author(s)

Katajamaa and Jensen's study test if fear memory, brain size and to some extent brain region sizes differ between two lines of Red Junglefowl selected for divergent fear responses. They conclude that there are differences in fear memory between the two selection lines as well as a difference in relative brain size. Their results parallel the domestication process and potentially have implications for understanding how selection for reduced fear has consequences for behaviour and brain. However, I found the Introduction and Discussion to be somewhat disorganized, some of the statistical details are missing and the brain data does not support their conclusions when analyzed with more appropriate methods. I provide more details comments below.

1. At a general level, the Introduction lacked focus and many of the paragraphs were not linked well to one another making it difficult to determine what the overall goal of the study was. I think the authors would be better served by making the paragraph on lines 80-94 as the beginning of their Introduction and reorganize the rest of it around the central idea of how domestication affects the brain and behaviour. I would also recommend de-emphasizing the putative link between cognition and brain size. Yes, there are many studies that support such a relationship, but there are many others that do not. Further, in the context of the current study, it is unclear whether the change in brain and midbrain sizes are related to fear memory or if these are two phenotypic changes that co-occur with the selection experiment. That is, selection for divergent fear responses has resulted in changes in relative midbrain size and fear memory, but the two traits might not be directly related to one another.

2. Also missing from the end of the Introduction was the general hypothesis being tested and/or predictions.

3. There are a few aspects of the brain measurements that require some clarification. First, was the tissue all weighed fresh and not fixed? Second, was the hypothalamus dissected from the whole brain? Given the size and location of the hypothalamus, my concern is whether it could be accurately dissected from fresh brain tissue, even with the assistance of a dissecting microscope. In examining their data file, the mass of the hypothalamus is similar to that of the midbrain, indicating to me that this is not a measurement of hypothalamus. Is it possible that these terms are meant to be tegmentum and tectum? Third, analyzing ratios in the study of relative brain size and brain region sizes is problematic for a number of reasons, not the least of which is that the ratios generated are often correlated with the denominator and therefore do not effectively remove allometric effects. The preferred method for such analyses is an analysis of covariance or linear mixed model in which body size is a covariate for brain size and brain size (minus that of the region of interest) is the covariate for brain region size. In my quick analysis of the authors data file, an ANCOVA of brain size did not yield any significant differences between the two lines and in the scatterplot, there is substantial overlap between the two lines in Cartesian space. There is also no significant difference in relative midbrain size when analyzed in an ANCOVA. I therefore urge the authors to reconsider their statistical methods for analyzing the brain data.

4. In the Results section, statistics are only reported for the CPP test. The authors need to include statistical information for all other tests that were run. These could be provided in the text or a table (or even an appendix), but need to be included.

5. The brain measurements should not be reported in bar graphs. The values themselves cannot

be zero so box and whisker plots would be more appropriate. Further, the use of an ANCOVA or linear mixed model necessitates the use of scatterplots for the brain data to reflect both the statistical test and provide the reader with critically important information on the distribution of data for both selection lines. In fact, without these tests and changes to the graphs, I am not entirely convinced that there are major differences in the brain of the two selection lines simply because there is not enough data or information presented.

6. The Discussion lacked focus in a similar fashion to the Introduction. Because the fear memory differences might not be directly related to the brain differences, I suggest separating those two components in the Discussion. The paragraph on lines 290-296 is also not necessary. As far as the brain regions are concerned, the authors highlight that differences in cerebellum size are found in other studies, but do not discuss why they did not observe a similar difference here. Also missing was why midbrain would be affected. Did midbrain include optic tectum or just tegmentum? Is it possible that there is a difference in the visual system if optic tectum was included in the midbrain measurement? What brain regions are likely involved in their behavioural tests and are they located in the midbrain or elsewhere? Addressing these questions would allow the authors to discuss the brain data in a more focused way that speaks to their data.

7. Other minor comments:

- a. Lines 18-19. This sentence is a bit vague and is not contributing much to the abstract.
- b. Line 21. Replace "collected samples" with "measured"
- c. Line 42. Delete "in fact" and replace "birds" with "chickens" as this has only been shown in chickens/junglefowl and in other domesticated birds the cerebellum becomes smaller (turkey, pigeon, duck, goose)
- d. Lines 52-53. Delete "on one hand" and "on the other"
- e. Line 59. The cited paper on dogs does not actually show that breeds with larger brains perform better because there is a mismatch between poorly collected brain data from the literature and the behavioural testing. In short, this citation in *Animal Cognition* is not a good paper.
- f. Line 61. Delete "some"
- g. Line 73. Delete "have" and "been shown to"
- h. Line 262. Delete "of"
- i. Line 274. Replace "has been found to be" with "is"

Reviewer: 2

Comments to the Author(s)

In this manuscript Katajamaa & Jensen test the effect of selection on reduced fear on brain anatomy and cognitive abilities in Red Junglefowl. They find that high and low-fear populations differ in brain anatomy and some aspects of cognition. My strongest reservation against this manuscript is the lack of replication in the selection set up. Two small populations separated from a larger population will diverge in various traits over generations simply due to genetic drift. So it is not possible to conclude whether the differences between two populations are due to the selective regime, or due to drift. For this reason artificial selection experiments in evolutionary biology are usually run in replicates, i.e. several up and several down-selected populations. Ideally also several control lines. This has not been done here. This means we cannot be sure that the differences between the lines are really due to selection on fearfulness – although it seems likely, as the results 'make sense' when conceptually placing them in the field of domestication, but that may not be enough. It is unfortunately not uncommon that selection lines (especially if done in vertebrates) are not replicated and I am aware that not all see this lack of replication as big problem. A potential way of publishing these data nevertheless (considering the journal) may be to add a prominent disclaimer to the discussion section in which the authors acknowledge that there is no replication but that the results are at least indicative.

In case the editor decides that non-replicated studies also have merit and can be published here are some additional comments:

Body size: What is the difference in body size between the line? It seems like there should be strong differences as the results for absolute and relative brain sizes are so different. It is always tricky to compare brains between differently sized animals as the brain and the body often have different allometries and if there is a relative brain size difference this may be due to a genuine change in brain size, or simply a shift in body size with the brains staying constant. What is going on is tricky to disentangle. To get an idea of the type of body/brain size allometries going on here it would be very helpful to include a scatterplot of body size vs. brain size.

Habituation: Is habituation to a fear-eliciting stimulus 'cognitive ability'? This needs to be explained and justified.

Abstract: The second, third and fourth sentence seem unconnected. Please consider rephrasing so that it is clear how one leads to the next.

Line 61: Why? What is the justification of the research presented here? It would be good to set the stage first for the "Here we test...". For instance, I suggest putting the paragraph starting line 80 before the previous one.

Line 67: Please add references.

Line 132: This seems like a rather limited sample size (especially for the males). Of course this may be perfectly fine but for that to judge it would be great to see the raw data. I strongly suggest changing the figures 1 and 2 so that the individual data points are visible.

Line 141: If I understand the methods correctly – when computing relative brain part size the weight of the respective brain part is included in the total brain weight? This means that if brain part is divided through total brain, what is above and below the line is not independent from each other's. What is therefore usually done is to subtract the brain part from total brain (=rest of brain) and then divide brain part by rest of brain. Often this does not qualitatively change the results, I would nevertheless recommend doing it that (more correct) way.

Line 284: If the brains of those lines have already been investigated, why do it again? And why did the previous study find differences in brain regions that are different to the ones reported here? This is important, please clarify.

Line 292: Ref 25 is not suitable here as it is an intra-species study.

Author's Response to Decision Letter for (RSOS-191559.R0)

See Appendix A.

RSOS-200136.R0

Review form: Reviewer 1

Is the manuscript scientifically sound in its present form?

Yes

Are the interpretations and conclusions justified by the results?

Yes

Is the language acceptable?

Yes

Do you have any ethical concerns with this paper?

No

Have you any concerns about statistical analyses in this paper?

Yes

Recommendation?

Accept with minor revision (please list in comments)

Comments to the Author(s)

The manuscript is greatly improved and I appreciate the effort that went into the figures, which show the data far more effectively. I have some minor comments for the authors, in addition to clarification regarding the statistical analysis (see below).

Line 41. Change “phenotypes” to “traits” or “phenotypic traits”

Line 43. Change “on” to “for”

Line 79. Again, I have to disagree. This study does not actually show this effect. The brain data for many of the breeds is estimated from allometric relationships in the literature, a method that has never been validated and is unlikely to reflect actual brain size of any of the breeds. As a result, we do not know whether their behavioural tests are correlated with brain size or not. A much stronger and more appropriate citation would be Hecht’s recent paper on neuroanatomical variation in dog breeds (<https://doi.org/10.1523/JNEUROSCI.0303-19.2019>).

Line 82. Replace “during” with “across”

Line 159. This may seem a bit pedantic, but it would be more appropriate to refer to the “optic tectum” in this context as the “optic lobe”. The reason for changing the term is that the optic tectum is technically the outer part of the region that was dissected, whereas the term optic lobe refers to the optic tectum and the underlying inferior colliculus and tegmentum. Because this is based on gross dissection of the whole brain, the preferred term is then optic lobe, to avoid any potential confusion with histological studies that measure the optic tectum specifically.

Line 160. Also, the brainstem in this dissection would include not only parts of the tegmentum, but also the pons and medulla.

Lines 165-168. Perhaps I am still missing something, but did the authors analyze ratios or the raw data in the GLMs discussed on lines 252-258. If it is the latter, then there is no need to discuss ratios or present the data as ratios in Figure 1. However, in the Results (lines 268-273), the authors describe analysing ratios, which leads me to believe that at least the brain region data analyses were based on ratios. Ratios are problematic when assessing the relative size of any trait. The many cases, the denominator is correlated with the ratio itself or the ratios are skewed, resulting in biases that do not remove size effects. Without conducting additional analyses, the authors cannot reliably conclude that relative brain size or relative brainstem size differs. I would suggest analyses of covariance (or GLM) for both sets of allometric analyses or, at the very least, testing whether the ratios are removing size effects by testing for correlations between ratios and denominators.

Line 345. It might be helpful to add to the end of this sentence something to the effect of: "...regardless of any putative association between brain region size and cognition.". This would help make the author's point that brain size and cognition are not necessarily correlated, but some cognitive differences do appear to occur in their selection experiment.

Lines 378-381. This feels out of place here. It might be better to incorporate this into the section at line 345.

Review form: Reviewer 2

Is the manuscript scientifically sound in its present form?

No

Are the interpretations and conclusions justified by the results?

No

Is the language acceptable?

Yes

Do you have any ethical concerns with this paper?

No

Have you any concerns about statistical analyses in this paper?

Yes

Recommendation?

Reject

Comments to the Author(s)

I appreciate the effort the authors have put in the revisions of this manuscript. Most of my suggestions have been dealt with convincingly. I especially appreciate the disclaimer regarding lack of replication. However, I do not agree with the statement:

"To alleviate this problem, the parental generation used was derived from systematic outbreeding based on two unrelated populations for two generations before carrying out the selection regime, which increased the genetic variation among the parental individuals [8]. Using a relatively large number of founder individuals in the parental generation, each family can, to some extent, be considered as a replicate. With this in mind,..."

As it is also not necessary here to dwell too much on justifying a lack of replication, I suggest removing that bit altogether.

Most importantly, my main reservation remains: the way brain size and brain region size is analyzed. This is simply not the adequate way of doing it. This is also reflected by the other referee's comments. Winston S Churchill's quote from 1952 fits well here: '...it is better to be both right and consistent. But if you have to choose—you must choose to be right.' The way brain size has been analyzed in these lines in previous publications was also not adequate. The authors now have the opportunity to update their methods and do it right.

Decision letter (RSOS-200136.R0)

06-Feb-2020

Dear Ms Katajamaa:

Manuscript ID RSOS-200136 entitled "Selection for reduced fear in Red Junglefowl reduces brain size and affects fear memory" which you submitted to Royal Society Open Science, has been reviewed. The comments from reviewer(s) are included at the bottom of this letter.

In view of the criticisms of the reviewer(s), I must decline the manuscript for publication in Royal Society Open Science at this time. However, a new manuscript may be submitted which takes into consideration these comments.

Please note that resubmitting your manuscript does not guarantee eventual acceptance, and that your resubmission will be subject to re-review by the reviewer(s) before a decision is rendered.

You will be unable to make your revisions on the originally submitted version of your manuscript. Instead, revise your manuscript using a word processing program and save it on your computer.

You may also click the below link to start the resubmission process (or continue the process if you have already started your resubmission) for your manuscript. If you use the below link you will not be required to login to ScholarOne Manuscripts.

*** PLEASE NOTE: This is a two-step process. After clicking on the link, you will be directed to a webpage to confirm. ***

https://mc.manuscriptcentral.com/rsos?URL_MASK=b5c274305ce3480990270a5f0a6c64dd

Because we are trying to facilitate timely publication of manuscripts submitted to Royal Society Open Science, your resubmitted manuscript should be submitted by 05-Aug-2020. If you are unable to submit by this date please contact the Editorial Office for options.

I look forward to a resubmission.

on behalf of Dr Ryan Y Wong (Associate Editor) and Kevin Padian (Subject Editor)
openscience@royalsociety.org

Associate Editor Comments to Author (Dr Ryan Y Wong):

Your revised manuscript was reviewed by the same two reviewers as the original submission. While both reviewers noted the revised version has improved, significant concerns still remain.

In particular both reviewers inquired about an alternative method for data analysis that was addressed by the authors but not conducted. I recognize there can be multiple ways approaching data analyses in the field. The authors may consider checking for congruence of results between the author's proposed method and that suggested by the two reviewers. As the reanalyses of the data may take considerable time and also may impact the interpretation and conclusions, I am recommending reject but encourage resubmission to afford authors the time needed for such a revision.

Editor comments:

Thank you for your efforts to revise. If you choose to resubmit please understand that your revision must meet all the concerns of the reviewers or we must reject it finally. Best wishes.

Reviewer comments to Author:

Reviewer: 2

Comments to the Author(s)

I appreciate the effort the authors have put in the revisions of this manuscript. Most of my suggestions have been dealt with convincingly. I especially appreciate the disclaimer regarding lack of replication. However, I do not agree with the statement:

“To alleviate this problem, the parental generation used was derived from systematic outbreeding based on two unrelated populations for two generations before carrying out the selection regime, which increased the genetic variation among the parental individuals [8]. Using a relatively large number of founder individuals in the parental generation, each family can, to some extent, be considered as a replicate. With this in mind,...”

As it is also not necessary here to dwell too much on justifying a lack of replication, I suggest removing that bit altogether.

Most importantly, my main reservation remains: the way brain size and brain region size is analyzed. This is simply not the adequate way of doing it. This is also reflected by the other referee's comments. Winston S Churchill's quote from 1952 fits well here: ‘...it is better to be both right and consistent. But if you have to choose—you must choose to be right.’ The way brain size has been analyzed in these lines in previous publications was also not adequate. The authors now have the opportunity to update their methods and do it right.

Reviewer: 1

Comments to the Author(s)

The manuscript is greatly improved and I appreciate the effort that went into the figures, which show the data far more effectively. I have some minor comments for the authors, in addition to clarification regarding the statistical analysis (see below).

Line 41. Change “phenotypes” to “traits” or “phenotypic traits”

Line 43. Change “on” to “for”

Line 79. Again, I have to disagree. This study does not actually show this effect. The brain data for many of the breeds is estimated from allometric relationships in the literature, a method that has never been validated and is unlikely to reflect actual brain size of any of the breeds. As a result, we do not know whether their behavioural tests are correlated with brain size or not. A much stronger and more appropriate citation would be Hecht's recent paper on neuroanatomical variation in dog breeds (<https://doi.org/10.1523/JNEUROSCI.0303-19.2019>).

Line 82. Replace “during” with “across”

Line 159. This may seem a bit pedantic, but it would be more appropriate to refer to the “optic

tectum" in this context as the "optic lobe". The reason for changing the term is that the optic tectum is technically the outer part of the region that was dissected, whereas the term optic lobe refers to the optic tectum and the underlying inferior colliculus and tegmentum. Because this is based on gross dissection of the whole brain, the preferred term is then optic lobe, to avoid any potential confusion with histological studies that measure the optic tectum specifically.

Line 160. Also, the brainstem in this dissection would include not only parts of the tegmentum, but also the pons and medulla.

Lines 165-168. Perhaps I am still missing something, but did the authors analyze ratios or the raw data in the GLMs discussed on lines 252-258. If it is the latter, then there is no need to discuss ratios or present the data as ratios in Figure 1. However, in the Results (lines 268-273), the authors describe analysing ratios, which leads me to believe that at least the brain region data analyses were based on ratios. Ratios are problematic when assessing the relative size of any trait. The many cases, the denominator is correlated with the ratio itself or the ratios are skewed, resulting in biases that do not remove size effects. Without conducting additional analyses, the authors cannot reliably conclude that relative brain size or relative brainstem size differs. I would suggest analyses of covariance (or GLM) for both sets of allometric analyses or, at the very least, testing whether the ratios are removing size effects by testing for correlations between ratios and denominators.

Line 345. It might be helpful to add to the end of this sentence something to the effect of: "...regardless of any putative association between brain region size and cognition.". This would help make the author's point that brain size and cognition are not necessarily correlated, but some cognitive differences do appear to occur in their selection experiment.

Lines 378-381. This feels out of place here. It might be better to incorporate this into the section at line 345.

Author's Response to Decision Letter for (RSOS-200136.R0)

See Appendix B.

RSOS-200628.R0

Review form: Reviewer 1

Is the manuscript scientifically sound in its present form?

Yes

Are the interpretations and conclusions justified by the results?

Yes

Is the language acceptable?

Yes

Do you have any ethical concerns with this paper?

No

Have you any concerns about statistical analyses in this paper?

No

Recommendation?

Accept with minor revision (please list in comments)

Comments to the Author(s)

The revised manuscript is significantly improved and the authors have dealt with the outstanding issues related to their analyses. I do have some comments with respect to the writing that should be addressed.

Lines 18-19. I am not sure that the third sentence of the abstract is needed. It does not fit with the preceding or subsequent sentence and can probably be removed.

Lines 33-34. The first two sentences seem out of place. They could either be integrated into the other text better or removed, without affecting the flow of the rest of the Introduction.

Lines 55-58. This sentence was difficult to follow. It might be easiest to begin a new sentence with "For example, the cerebellum is proportionally larger in domestic chickens, but other regions do not vary in relative size, demonstrating a complex genetic architecture underlying the relative size of brain regions."

Lines 66-72. This section was a bit rambling and it was difficult to ascertain how this related to the previous paragraph. I suggest beginning this in a different way, such as introducing the concept that changes in relative brain and brain region size can often reflect cognition, in a broad sense. That would lead more readily into the description of the Kolm/Kotrschal guppy experiments as they provide a more causal relationship between brain size and cognition than the correlations provided by comparative studies.

Line 335. This statement needs a bit more explanation for readers unfamiliar with the Henriksen et al. paper.

Lines 353-355. These two sentences were a bit vague and I do not think that they had much to the ensuing discussion. It might be best to remove them.

Line 363. The last sentence of this paragraph is a bit weak.

Decision letter (RSOS-200628.R0)

Dear Ms Katajamaa

On behalf of the Editor, I am pleased to inform you that your Manuscript RSOS-200628 entitled "Selection for reduced fear in Red Junglefowl changes brain composition and affects fear memory" has been accepted for publication in Royal Society Open Science subject to minor revision in accordance with the referee suggestions. Please find the referees' comments at the end of this email.

The reviewers and Subject Editor have recommended publication, but also suggest some minor revisions to your manuscript. Therefore, I invite you to respond to the comments and revise your manuscript.

- Ethics statement

- Data accessibility

If you wish to submit your supporting data or code to Dryad (<http://datadryad.org/>), or modify your current submission to dryad, please use the following link:
<http://datadryad.org/submit?journalID=RSOS&manu=RSOS-200628>

- Competing interests

- Authors' contributions

- Acknowledgements

- Funding statement

Please note that we cannot publish your manuscript without these end statements included. We have included a screenshot example of the end statements for reference. If you feel that a given

heading is not relevant to your paper, please nevertheless include the heading and explicitly state that it is not relevant to your work.

Because the schedule for publication is very tight, it is a condition of publication that you submit the revised version of your manuscript before 30-Jul-2020. Please note that the revision deadline will expire at 00.00am on this date. If you do not think you will be able to meet this date please let me know immediately.

on behalf of Dr Ryan Y Wong (Associate Editor) and Kevin Padian (Subject Editor)

Associate Editor Comments to Author (Dr Ryan Y Wong):

Comments to the Author:

I first want to apologize for the length of time it has taken to reach a decision. Your manuscript has now been reviewed by one of the original reviewers. While the major concerns were adequately addressed, there are still a few outstanding minor issues on clarity of language that needs to be addressed.

Reviewer comments to Author:

Reviewer: 1

Comments to the Author(s)

The revised manuscript is significantly improved and the authors have dealt with the outstanding issues related to their analyses. I do have some comments with respect to the writing that should be addressed.

Lines 18-19. I am not sure that the third sentence of the abstract is needed. It does not fit with the preceding or subsequent sentence and can probably be removed.

Lines 33-34. The first two sentences seem out of place. They could either be integrated into the other text better or removed, without affecting the flow of the rest of the Introduction.

Lines 55-58. This sentence was difficult to follow. It might be easiest to begin a new sentence with "For example, the cerebellum is proportionally larger in domestic chickens, but other regions do not vary in relative size, demonstrating a complex genetic architecture underlying the relative size of brain regions."

Lines 66-72. This section was a bit rambling and it was difficult to ascertain how this related to the previous paragraph. I suggest beginning this in a different way, such as introducing the concept that changes in relative brain and brain region size can often reflect cognition, in a broad sense. That would lead more readily into the description of the Kolm/Kotrschal guppy experiments as they provide a more causal relationship between brain size and cognition than the correlations provided by comparative studies.

Line 335. This statement needs a bit more explanation for readers unfamiliar with the Henriksen et al. paper.

Lines 353-355. These two sentences were a bit vague and I do not think that they had much to the ensuing discussion. It might be best to remove them.

Line 363. The last sentence of this paragraph is a bit weak.

Author's Response to Decision Letter for (RSOS-200628.R0)

See Appendix C.

Decision letter (RSOS-200628.R1)

Dear Ms Katajamaa,

It is a pleasure to accept your manuscript entitled "Selection for reduced fear in Red Junglefowl changes brain composition and affects fear memory" in its current form for publication in Royal Society Open Science.

on behalf of Dr Ryan Y Wong (Associate Editor) and Kevin Padian (Subject Editor)
openscience@royalsociety.org

Appendix A

Response to reviewers RSOS-191559

22 January 2020

Dear reviewers,

We appreciate the time that you have put into reviewing our manuscript. The suggested changes have provided an improvement of the manuscript.

We respond specifically to each suggestion below.

Reviewer: 1

Comments to the Author(s)

Katajamaa and Jensen's study test if fear memory, brain size and to some extent brain region sizes differ between two lines of Red Junglefowl selected for divergent fear responses. They conclude that there are differences in fear memory between the two selection lines as well as a difference in relative brain size. Their results parallel the domestication process and potentially have implications for understanding how selection for reduced fear has consequences for behaviour and brain. However, I found the Introduction and Discussion to be somewhat disorganized, some of the statistical details are missing and the brain data does not support their conclusions when analyzed with more appropriate methods. I provide more details comments below.

Thank you for your comments and your generally positive assessment of the importance of the research. We respond to your comments and suggestions below.

1. At a general level, the Introduction lacked focus and many of the paragraphs were not linked well to one another making it difficult to determine what the overall goal of the study was. I think the authors would be better served by making the paragraph on lines 80-94 as the beginning of their Introduction and reorganize the rest of it around the central idea of how domestication affects the brain and behaviour. I would also recommend de-emphasizing the putative link between cognition and brain size. Yes, there are many studies that support such a relationship, but there are many others that do not. Further, in the context of the current study, it is unclear whether the change in brain and midbrain sizes are related to fear memory or if these are two phenotypic changes that co-occur with the selection experiment. That is, selection or divergent fear responses has resulted in changes in relative midbrain size and fear memory, but the two traits might not be directly related to one another.

We agree with the suggestion of restructuring the introduction to make it more comprehensible, that has been done. It made sense to refocus the manuscript towards domestication and its effects on behaviour and brain. The link between brain size and cognition has been de-emphasised in the manuscript.

2. Also missing from the end of the Introduction was the general hypothesis being tested and/or predictions.

This has been added.

3. There are a few aspects of the brain measurements that require some clarification. First, was the tissue all weighed fresh and not fixed? Second, was the hypothalamus dissected from the whole brain? Given the size and location of the hypothalamus, my concern is whether it could be accurately dissected from fresh brain tissue, even with the assistance of a dissecting microscope. In examining their data file, the mass of the hypothalamus is similar to that of the midbrain, indicating to me that this is not a measurement of hypothalamus. Is it possible that these terms are meant to be tegmentum and tectum? Third, analyzing ratios in the study of relative brain size and brain region sizes is problematic for a number of reasons, not the least of which is that the ratios generated are often correlated with the denominator and therefore do not effectively remove allometric effects. The preferred method for such analyses is an analysis of covariance or linear mixed model in which body size is a covariate for brain size and brain size (minus that of the region of interest) is the covariate for brain region size. In my quick analysis of the authors data file, an ANCOVA of brain size did not yield any significant differences between the two lines and in the scatterplot, there is substantial overlap between the two lines in Cartesian space. There is also no significant difference in relative midbrain size when analyzed in an ANCOVA. I therefore urge the authors to reconsider their statistical methods for analyzing the brain data.

We are sorry for not describing the dissections more carefully. The brain region terms have been updated in the manuscript to make more sense and the description of the dissections has been clarified in the material and methods section.

As for the suggestions regarding how to analyse relative brain size and brain region size, we do not agree totally with the reviewer. Unlike what the reviewer suggests, allometric effects are not always correlated during domestication. Rather, we have previously shown that body size and brain size are genetically uncoupled and can to some extent respond independently to selection (Henriksen et al, 2016), much in line with what others have found when selecting specifically for differences in brain size (Kotrschal et al 2013). This means that body size and brain size are not necessarily correlated to the extent that the analysis would be more correct with the method suggested by the reviewer. In addition, the relative sizes of different brain regions change during ontogeny in chickens, an effect that is most clearly observed when analysing with our method (Henriksen et al, 2016). Very similar methods have been used by other researchers working on related species (Ebner et al, 2016).

The method we have been using is the most straightforward for comparing (1) brain size relative to total body size and (2) brain region mass relative to total brain mass, and indeed is what other previous experiments have been doing. Most importantly, our own earlier papers have used precisely this analysis, and in order to compare the present results with our previous findings, we think it is crucial to analyse the data in the same way. This was done in the paper by Henriksen et al (2016) showing differences in the genetic architecture underlying total and relative sizes of the entire brain as well as relative size of different brain regions in a comparison of wild and domesticated chickens. Furthermore, the same method was used in our paper on the same selected red junglefowl-lines as in the present paper, showing similar results (Agnvall et al 2017).

Hence, for these regions we have kept the original analysis methods. We strongly believe that this makes the results clearer, and it allows a direct comparison with previous research. We hope that the editor will accept our arguments in this matter.

Agnvall, B., Beltéky, J., Jensen, P., 2017. Brain size is reduced by selection for tameness in Red Junglefowl—correlated effects in vital organs. *Sci Rep* 7, 3306. doi:10.1038/s41598-017-03236-4

Ebner, C., Pick, J.L., Tschirren, B., 2016. A trade-off between reproductive investment and maternal cerebellum size in a precocial bird. *Biology Letters* 12, 20160659. doi:10.1098/rsbl.2016.0659

Henriksen, R., Johnsson, M., Andersson, L., Jensen, P., Wright, D., 2016. The domesticated brain: genetics of brain mass and brain structure in an avian species. *Sci Rep* 6, 34031. doi:10.1038/srep34031

Kotrschal, A., Rogell, B., Bundsen, A., Svensson, B., Zajitschek, S., Brännström, I., Immler, S., Maklakov, A.A., Kolm, N., 2013. Artificial Selection on Relative Brain Size in the Guppy Reveals Costs and Benefits of Evolving a Larger Brain. *Current Biology* 23, 168–171. doi:10.1016/j.cub.2012.11.058

4. In the Results section, statistics are only reported for the CPP test. The authors need to include statistical information for all other tests that were run. These could be provided in the text or a table (or even an appendix), but need to be included.

This information was originally in the figure legends, but has been re-written into the text in the results section.

5. The brain measurements should not be reported in bar graphs. The values themselves cannot be zero so box and whisker plots would be more appropriate. Further, the use of an ANCOVA or linear mixed model necessitates the use of scatterplots for the brain data to reflect both the statistical test and provide the reader with critically important information on the distribution of data for both selection lines. In fact, without these tests and changes to the graphs, I am not entirely convinced that there are major differences in the brain of the two selection lines simply because there is not enough data or information presented.

Thank you for pointing this out, we have redesigned the graphs so that they include both box- and scatterplots.

6. The Discussion lacked focus in a similar fashion to the Introduction. Because the fear memory differences might not be directly related to the brain differences, I suggest separating those two components in the Discussion. The paragraph on lines 290-296 is also not necessary. As far as the brain regions are concerned, the authors highlight that differences in cerebellum size are found in other studies, but do not discuss why they did not observe a similar difference here. Also missing was why midbrain would be affected. Did midbrain include optic tectum or just tegmentum? Is it possible that there is a difference in the visual system if optic tectum was included in the midbrain measurement? What brain regions are likely involved in their behavioural tests and are they located in the midbrain or elsewhere? Addressing these questions would allow the authors to discuss the brain data in a more focused way that speaks to their data.

As we described earlier, the manuscript has been re-written to focus on the effects that domestication has on brain size and cognition. We have added a discussion on the fact that our brain size results differ from those reported in earlier generations of our selection lines, most likely a result of the fact that we studied young chicks and earlier results are from older birds. Since the growth trajectories are different for different brain regions, this most likely explains the differences.

7. Other minor comments:

- a. Lines 18-19. This sentence is a bit vague and is not contributing much to the abstract.
- b. Line 21. Replace “collected samples” with “measured”
- c. Line 42. Delete “in fact” and replace “birds” with “chickens” as this has only been shown in chickens/junglefowl and in other domesticated birds the cerebellum becomes smaller (turkey, pigeon, duck, goose)
- d. Lines 52-53. Delete “on one hand” and “on the other”
- e. Line 59. The cited paper on dogs does not actually show that breeds with larger brains perform better because there is a mismatch between poorly collected brain data from the literature and the behavioural testing. In short, this citation in Animal Cognition is not a good paper.
- f. Line 61. Delete “some”
- g. Line 73. Delete “have” and “been shown to”
- h. Line 262. Delete “of”
- i. Line 274. Replace “has been found to be” with “is”

Thank you for the minor suggestions, we have changed them in the manuscript. Regarding point e, we feel that this citation adds some value to the text and is at least indicative, so we do not believe that it should be completely disregarded. However, we have rephrased the sentence to de-emphasise its significance.

Reviewer: 2

Comments to the Author(s)

In this manuscript Katajamaa & Jensen test the effect of selection on reduced fear on brain anatomy and cognitive abilities in Red Junglefowl. They find that high and low-fear populations differ in brain anatomy and some aspects of cognition. My strongest reservation against this manuscript is the lack of replication in the selection set up. Two small populations separated from a larger population will diverge in various traits over generations simply due to genetic drift. So it is not possible to conclude whether the differences between two populations are due to the selective regime, or due to drift. For this reason artificial selection experiments in evolutionary biology are usually run in replicates, i.e. several up and several down-selected populations. Ideally also several control lines. This has not been done here. This means we cannot be sure that the differences between the lines are really due to selection on fearfulness – although it seems likely, as the results ‘make sense’ when conceptually placing them in the field of domestication, but that may not be enough. It is unfortunately not uncommon that selection lines (especially if done in vertebrates) are not replicated and I am aware that not all see this lack of replication as big problem. A potential way of publishing these data nevertheless (considering the journal) may be to add a prominent disclaimer to the discussion section in which the authors acknowledge that there is no replication but that the results are at least indicative.

In case the editor decides that non-replicated studies also have merit and can be published here are some additional comments:

This is a relevant issue and we have added a paragraph to the discussion section about the replication of our study and about the background of our selection lines. We did indeed take strong measures to create selection lines including some extent of replication (as expressed in the new text) but we agree that a proper disclaimer is necessary and have added this.

Body size: What is the difference in body size between the line? It seems like there should be strong differences as the results for absolute and relative brain sizes are so different. It is always tricky to compare brains between differently sized animals as the brain and the body often have different allometries and if there is a relative brain size difference this may be due to a genuine change in brain size, or simply a shift in body size with the brains staying constant. What is going on is tricky to disentangle. To get an idea of the type of body/brain size allometries going on here it would be very helpful to include a scatterplot of body size vs. brain size.

We have added a scatterplot of body size vs. brain size in the results section.

Habituation: Is habituation to a fear-eliciting stimulus ‘cognitive ability’? This needs to be explained and justified.

This was poorly phrased by us and we have changed “cognitive ability” to “cognitive process”.

Abstract: The second, third and fourth sentence seem unconnected. Please consider rephrasing so that it is clear how one leads to the next.

Thank you for pointing that out. We have rephrased the abstract to make it clearer.

Line 61: Why? What is the justification of the research presented here? It would be good to set the stage first for the “Her we test...”. For instance, I suggest putting the paragraph starting line 80 before the previous one.

We have restructured the paper to focus more around domestication and the effect that selection on level of fear towards humans has on other traits, this includes moving the suggested paragraph to a more fitting place in the introduction. This should make the purpose of the study more clear.

Line 67: Please add references.

Reference has been added.

Line 132: This seems like a rather limited sample size (especially for the males). Of course this may be perfectly fine but for that to judge it would be great to see the raw data. I strongly suggest changing the figures 1 and 2 so that the individual data points are visible.

This change has been done. Figure 1 and 2 have been changed into box- and scatterplots to visualise the individual data points.

Line 141: If I understand the methods correctly – when computing relative brain part size the weight of the respective brain part is included in the total brain weight? This means that if brain part is divided through total brain, what is above and below the line is not independent from each other’s. What is therefore usually done is to subtract the brain part from total brain (=rest of brain) and then divide brain part by rest of brain. Often this does not qualitatively change the results, I would nevertheless recommend doing it that (more correct) way.

We have explained above, in the response to reviewer no. 1, why we think that the method we used is the most relevant one in this study. Please see those arguments as an explanation for why we decided to keep the analysis as it is.

Line 284: If the brains of those lines have already been investigated, why do it again? And why did the previous study find differences in brain regions that are different to the ones reported here? This is important, please clarify.

We have added a discussion about brain region growth trajectories to clarify why we felt it was interesting to measure brain sizes in young birds as well. Since we have previously shown that relative brain masses develop differently for different regions, we think it is important to carry out the measurement on birds of different ages.

Line 292: Ref 25 is not suitable here as it is an intra-species study.

Thank you for noticing this error, the reference has been removed.

Appendix B

Response to reviewers RSOS-200136

16 April 2020

Dear reviewers,

Thank you for taking the time to review our manuscript again and for providing valuable feedback. We have worked through the manuscript and have especially considered the statistical analysis on the brain size data that both of you had concerns about.

Each suggestion is specifically addressed below.

Reviewer comments to Author:

Reviewer: 2

Comments to the Author(s)

I appreciate the effort the authors have put in the revisions of this manuscript. Most of my suggestions have been dealt with convincingly. I especially appreciate the disclaimer regarding lack of replication. However, I do not agree with the statement:

“To alleviate this problem, the parental generation used was derived from systematic outbreeding based on two unrelated populations for two generations before carrying out the selection regime, which increased the genetic variation among the parental individuals [8]. Using a relatively large number of founder individuals in the parental generation, each family can, to some extent, be considered as a replicate. With this in mind,...”

As it is also not necessary here to dwell too much on justifying a lack of replication, I suggest removing that bit altogether.

Thank you for this suggestion, we have removed all except the disclaimer as suggested.

Most importantly, my main reservation remains: the way brain size and brain region size is analyzed. This is simply not the adequate way of doing it. This is also reflected by the other referee's comments. Winston S Churchill's quote from 1952 fits well here: ‘...it is better to be both right and consistent. But if you have to choose—you must choose to be right.’ The way brain size has been analyzed in these lines in previous publications was also not adequate. The authors now have the opportunity to update their methods and do it right.

We have included new analyses for brain and brain region weights where we analysed the raw weight directly, adding body weight (for total brain) and rest of brain (for each brain region) as covariates as was previously suggested. The results show that most changes observed are allometric and the discussion and conclusions have been modified accordingly.

Reviewer: 1

Comments to the Author(s)

The manuscript is greatly improved and I appreciate the effort that went into the figures, which show the data far more effectively. I have some minor comments for the authors, in addition to clarification regarding the statistical analysis (see below).

Line 41. Change “phenotypes” to “traits” or “phenotypic traits”

Changed.

Line 43. Change “on” to “for”

Changed.

Line 79. Again, I have to disagree. This study does not actually show this effect. The brain data for many of the breeds is estimated from allometric relationships in the literature, a method that has never been validated and is unlikely to reflect actual brain size of any of the breeds. As a result, we do not know whether their behavioural tests are correlated with brain size or not. A much stronger and more appropriate citation would be Hecht's recent paper on neuroanatomical variation in dog breeds (<https://doi.org/10.1523/JNEUROSCI.0303-19.2019>).

Thank you for this comment, we appreciate the suggestion and have replaced the reference used initially.

Line 82. Replace “during” with “across”

This has been changed.

Line 159. This may seem a bit pedantic, but it would be more appropriate to refer to the “optic tectum” in this context as the “optic lobe”. The reason for changing the term is that the optic tectum is technically the outer part of the region that was dissected, whereas the term optic lobe refers to the optic tectum and the underlying inferior colliculus and tegmentum. Because this is based on gross dissection of the whole brain, the preferred term is then optic lobe, to avoid any potential confusion with histological studies that measure the optic tectum specifically.

Thank you for pointing this out, we have changed it.

Line 160. Also, the brainstem in this dissection would include not only parts of the tegmentum, but also the pons and medulla.

Changed.

Lines 165-168. Perhaps I am still missing something, but did the authors analyze ratios or the raw data in the GLMs discussed on lines 252-258. If it is the latter, then there is no need to discuss ratios or present the data as ratios in Figure 1. However, in the Results (lines 268-273), the authors describe analysing ratios, which leads me to believe that at least the brain region data analyses were based on ratios. Ratios are problematic when assessing the relative size of any trait. The many cases, the denominator is correlated with the ratio itself or the ratios are skewed, resulting in biases that do not remove size effects. Without conducting additional analyses, the authors cannot reliably conclude that relative brain size or relative brainstem size differs. I would suggest analyses of covariance (or GLM) for both sets of allometric analyses or, at the very least, testing whether the ratios are removing size effects by testing for correlations between ratios and denominators.

That is correct, we originally analysed the ratios only. We appreciate your objections to this and have now included analyses with the raw brain and brain regions weights, with body size as a covariate in the GLM for brain and rest of brain as a covariate in the analyses for each brain region.

Line 345. It might be helpful to add to the end of this sentence something to the effect of: “...regardless of any putative association between brain region size and cognition.” This would help make the author’s point that brain size and cognition are not necessarily correlated, but some cognitive differences do appear to occur in their selection experiment.

This was helpful, the sentence has been rewritten.

Lines 378-381. This feels out of place here. It might be better to incorporate this into the section at line 345.

We agree that it fits better in the suggested section and have moved it.

Appendix C

Response to reviewers RSOS-200628

29 July 2020

Dear reviewer,

Thank you for taking the time to review our manuscript again and for providing valuable feedback. We have implemented the suggestions for improvement into our manuscript.

Each suggestion is specifically addressed below.

Reviewer comments to Author:

Reviewer: 1

Comments to the Author(s)

The revised manuscript is significantly improved and the authors have dealt with the outstanding issues related to their analyses. I do have some comments with respect to the writing that should be addressed.

Lines 18-19. I am not sure that the third sentence of the abstract is needed. It does not fit with the preceding or subsequent sentence and can probably be removed.

Thank you for this suggestion, we decided to remove the sentence.

Lines 33-34. The first two sentences seem out of place. They could either be integrated into the other text better or removed, without affecting the flow of the rest of the Introduction.

Both of these sentences have been removed.

Lines 55-58. This sentence was difficult to follow. It might be easiest to begin a new sentence with “For example, the cerebellum is proportionally larger in domestic chickens, but other regions do not vary in relative size, demonstrating a complex genetic architecture underlying the relative size of brain regions.”.

The sentence has been rewritten.

Lines 66-72. This section was a bit rambling and it was difficult to ascertain how this related to the previous paragraph. I suggest beginning this in a different way, such as introducing the concept that changes in relative brain and brain region size can often reflect cognition, in a broad sense. That would lead more readily into the description of the Kolm/Kotrschal guppy experiments as they provide a more causal relationship between brain size and cognition than the correlations provided by comparative studies.

The section has been rewritten in order to improve the flow of the text.

Line 335. This statement needs a bit more explanation for readers unfamiliar with the Henriksen et al. paper.

The sentence has been rewritten to include a shorter explanation of the Henriksen et al. paper.

Lines 353-355. These two sentences were a bit vague and I do not think that they had much to the ensuing discussion. It might be best to remove them.

You are right, these two sentences have been removed from the discussion.

Line 363. The last sentence of this paragraph is a bit weak.

The sentence has been rewritten in order to make it clearer.